

# A scatterometer record of sea ice extents and backscatter: 1992-2016

Maria Belmonte Rivas[1,3], Ines Otosaka[1,2], Ad Stoffelen[1], Anton Verhoef[1]

[1]Royal Netherlands Meteorology Institute (KNMI), de Bilt, 3731 GA, The Netherlands
[2]Center for Polar Observation and Modelling (CPOM), University of Leeds, Leeds, LS2 9JT, United Kingdom
[3]Instituto de Ciencias del Mar (ICM), Consejo Superior de Investigaciones Cientificas (CSIC), 08003, Barcelona, Spain

*Correspondence to*: Maria Belmonte Rivas (belmonte@knmi.nl)

**Abstract.** This paper presents the first long-term climate data record of sea ice extents and backscatter derived from inter-calibrated satellite scatterometer missions (ERS, QuikSCAT and ASCAT) extending from 1992 to present date. This record provides a valuable independent account of the evolution of Arctic and Antarctic sea ice extents, one that is in excellent
agreement with the passive microwave records during the fall and winter months but shows higher sensitivity to lower concentration and melting sea ice during the spring and summer months, providing a means to correct for summer melt ponding errors. The scatterometer record also provides a depiction of sea ice backscatter at C and Ku-band, allowing the separation of seasonal and perennial sea ice in the Arctic, and further differentiation between second year (SY) and older multiyear (MY) ice classes, revealing the emergence of SY ice as the dominant perennial ice type after the record sea ice loss
in 2007, and bearing new evidence on the loss of multiyear ice in the Arctic over the last 25 years. The relative good agreement between the backscatter-based sea ice (FY, SY and older MY) classes and the ice thickness record from Cryosat suggests its applicability as a reliable proxy in the historical reconstruction of sea ice thickness in the Arctic.

## 1 Introduction

Dating as far back as 1979, passive microwave sensors provide the longest record of sea ice concentration and extents
available to date, and are currently established as the sea ice monitoring standard for climate studies, regardless of well-known difficulties around the detection of lower concentration and melting sea ice conditions during the summer months (Meier et al., 2015). The scatterometer sea ice record presented here only dates back as far as 1992, but proves more sensitive to summer sea ice, its primary purpose being the conservative detection and removal of ice contaminated scenes that compromise scatterometer wind retrievals. Previous long-term scatterometer sea ice records have been developed
spanning the decade-long QuikSCAT mission from 1999 to 2009 (Remund and Long, 2014), and extended into 2014 using the Oceansat-2 scatterometer (OSCAT) mission (Hill and Long, 2017). These precedent scatterometer records (which use maximum likelihood class discrimination based on the Ku-band pseudo-polarization HH/VV ratio and other parameters) already underline the presence of negative biases in passive microwave sea ice extents during the melt season, but also feature instances of missing thin ice during the growth season (Meier and Stroeve, 2008).





Some research groups have opted to blend active and passive microwave observations (i.e. the gradient and polarization ratios from radiometer, along with the C-band anisotropy coefficient from scatterometer data) in a multi-sensor approach towards a sea ice edge product (Aaboe, Breivik and Eastwood, 2015). However, sea ice extents from blended records are still negatively biased in the summer relative to operational sea ice charts by up to 30% (Aaboe et al., 2016), indicating that the

distinct sea ice detection skills of scatterometer data may be lost in the blend.

In this paper, an independent record of sea ice extents has been produced from inter-calibrated scatterometer data: the QuikSCAT mission from 1999 to 2009 (Belmonte Rivas and Stoffelen, 2011), extended forward to present day with the ASCAT record (Belmonte Rivas et al., 2012), and backwards to 1992 with the ERS mission (Otosaka et al., 2017), using

dedicated Bayesian sea ice detection algorithms designed to maximize the skill for ocean/ice discrimination. These algorithms have been tuned to match the passive microwave sea ice extents during the fall and winter months, and to remain consistent across the scatterometer overlap periods in 2000 (ERS with QuikSCAT) and 2008 (QuikSCAT with ASCAT). The stability and inter-calibration of the ERS, QuikSCAT and ASCAT backscatter records is guaranteed to within 0.1 dB via buoy collocation (QuikSCAT; Verhoef and Stoffelen, 2016), ocean calibration (ASCAT and ERS; Verhoef and Stoffelen,

2017) and nonlinear corrections from cone metrics (ERS; Belmonte Rivas et al, 2017), offering a stable reference to verify the consistency of calibration adjustments made in passive microwave records, which are known to cause discontinuities and affect long-term trends in sea ice concentration (Eisenman, Meier and Norris, 2014) (Titchner and Rayner, 2014).

The scatterometer sea ice record also monitors the evolution of sea ice backscatter collected at C-band and Ku-band, which

are widely applied to discriminate ice classes, such as first year (FY) and older (second year SY, and multiyear MY) sea ice in the Arctic. It is known that sea ice backscatter is modulated by surface permittivity, surface roughness and the presence of volume inhomogeneities, such as air and brine pockets, or snow layers above. The main basis for FY/MY ice separation lies in older ice types becoming brighter with increased volume scattering after summer melt, although MY detection may become difficult by patches of bright FY ice that may have locally undergone deformation. To date, the separation between

Arctic FY and MY ice types using active microwaves has relied on fixed backscatter thresholds defined after visual inspection of stable winter backscatter histograms. For example, (Kwok, 2004) established -14.5 dB as an optimal threshold for the QuikSCAT Ku-band VV polarized measurements based on visual examination of the subjective FY and MY ice boundaries in combined winter data sets of QuikSCAT and C-band SAR from RADARSAT. Other than calibration issues, the main problem with the fixed-threshold approach is the seasonal variability of the FY-MY backscatter signatures, along

with sensitivity to deformed FY ice types or a developing snow cover. On the other hand, the classification of sea ice types using passive microwaves (Gloersen and Cavalieri, 1986; Comiso, 2012) has relied on the spectral gradient and polarization signatures of sea ice brightness temperatures (with MY surfaces featuring more negative spectral gradients and lower polarization than FY ice, along with lower emissivities). The spatial and temporal distributions of perennial ice derived from passive microwaves in the Arctic have been shown to differ somewhat from those of SAR (Kwok, Comiso and Cunningham,





1996), their differences depending on atmospheric conditions and processes that affect the ice temperature and emissivity in ways that contribute to apparent concentration changes (Thomas, 1993). According to the IPCC AR5, the rate of decline in the extent of multiyear ice observed by both passive and active microwaves is consistent with the decline of old ice types estimated from the analysis of ice drift by (Maslanik et al., 2007), confirming that the Arctic has lost much of its thicker ice.

Still differences remain between scatterometer and radiometer multiyear ice extents (Comiso, 2012) associated to different sensitivities to sea ice type and snow cover, which should be better understood.

The introduction of an inter-calibrated sea ice extent and backscatter record from multiple scatterometer missions (ERS, QuikSCAT, ASCAT) consistently connected from 1992 to 2016 through dedicated and validated sea ice detection and

backscatter normalization algorithms is the object of this contribution. In Section 2, we introduce the satellite scatterometer missions and the Bayesian detection algorithms that constitute this record. In Section 3, the scatterometer sea ice extents are compared against passive microwave fields, showcasing their agreement and distinct sensitivities. This section also provides an overview of the long-term evolution of sea ice extents and sea ice types afforded by 25 years of scatterometer data, along with a taste of its potential to stimulate new research questions.

## 15 2 Satellite scatterometer missions

The continuous monitoring of the Polar Regions with satellite scatterometers began in March 1992 after the launch of the European Remote Sensing (ERS) satellites, which operated a C-band instrument (5.3 GHz, VV polarization) in global mode until January 2001. It was continued on July 1999 by the Quick Scatterometer (QuikSCAT), which operated a Ku-band instrument (13.4 GHz, HH and VV polarization) up until November 2009. The scatterometer record ends with another C-

band instrument that collects VV polarized backscatter at 5.3 GHz, the Advanced Scatterometer (ASCAT), operating from January 2007 to the present date. The temporal spans of the satellite missions that make up the scatterometer record are illustrated in Fig. 1, showing the location of the mission transition periods (2000 and 2008) used to verify the consistency of the sea ice extents and sea ice types across the C and Ku-band missions. For reference, the observation geometries of the constitutive ERS, QuikSCAT and ASCAT scatterometers (namely, three single-sided VV polarization fan-beams, single

rotating VV and HH polarization pencil beams, and three double-sided fan-beams, respectively) are shown in Fig. 2.



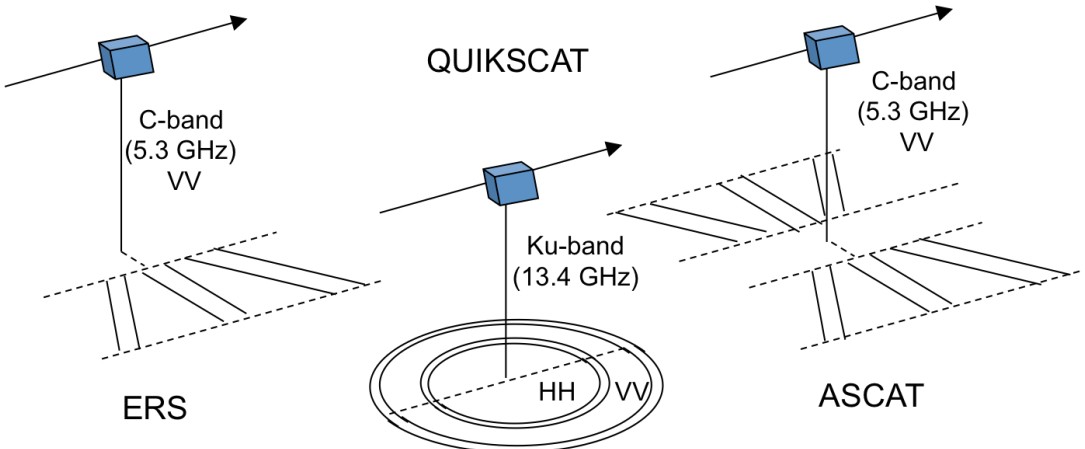

**Figure 1: Temporal spans of the satellite scatterometer missions used.**

**Figure 2: Observation geometries of the satellite scatterometers.**

### 2.1 Sea ice detection with scatterometers

The algorithm for sea ice detection with scatterometers is a maximum likelihood class discrimination approach based on probabilistic distances to ocean wind and sea ice geophysical model functions (GMFs). The GMFs describe the behaviour of backscatter as a function of observation geometry (i.e., incidence and azimuth angles) and geophysical variables such as

10 wind speed and direction, or sea ice type.




### 2.2 Geophysical model functions

The ocean wind GMF, also known as the wind cone, is an empirically derived model used to derive ocean surface wind vectors operationally (see Fig. 3): we presently use CMOD7.1 (Stoffelen et al., 2017) for ERS and ASCAT, and NSCAT-4 (Verhoef and Stoffelen, 2017) for QuikSCAT. The sea ice GMF, also known as the sea ice line, is empirically derived from

5 stable wintertime backscatter levels (Belmonte Rivas and Stoffelen, 2011; Belmonte Rivas et al., 2012; Otosaka et al., 2017). Physically, the discrimination between open water and sea ice classes is based on the separability between surface and volume scattering effects: in the QuikSCAT case, the discrimination relies on polarization and azimuthal anisotropy of backscatter (high for open water; lower for sea ice), while in the ERS/ASCAT case, the discrimination relies on backscatter directivity (i.e. the derivative of backscatter with incidence angle) and azimuthal anisotropy (high for open water; lower for

sea ice). Previous Bayesian formulations for sea ice detection with scatterometer data, e.g., (Remund and Long, 2014), have used aggregates such as mean backscatter, polarization ratio and azimuthal anisotropy as class discriminants, and empirically adjusted covariances to represent the class dispersions. The advantage of the GMF approach is that the dispersion of measurements about extended class model functions is smaller than about class aggregate means, approaching the limits imposed by the scatterometer noise levels, and allowing the Bayesian method to reach its maximum discrimination power

(Otosaka et al., 2017).

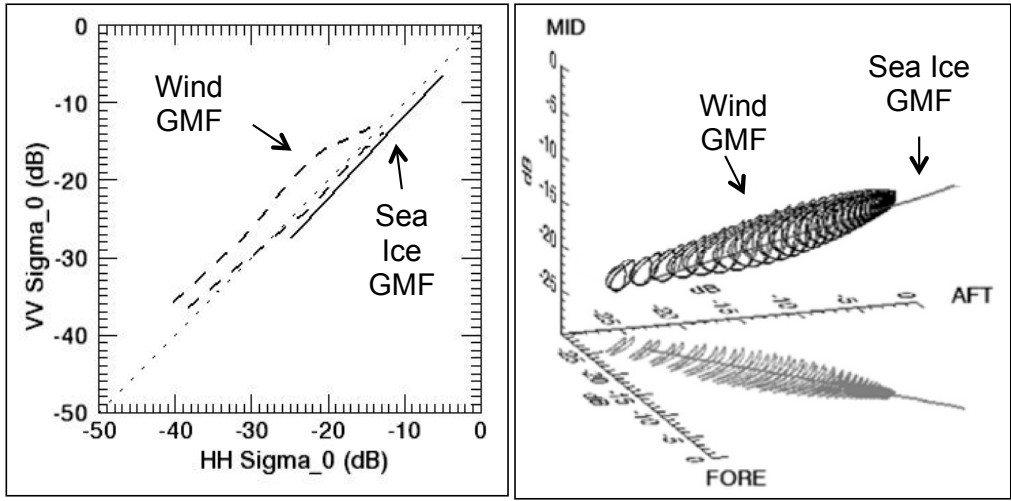

**Figure 3: Ocean wind and sea ice Geophysical Model Functions (GMFs) in the scatterometer measurement space: QuikSCAT (left) and ERS/ASCAT (right). Plots adapted from (Belmonte Rivas and Stoffelen, 2011; Belmonte Rivas et al., 2012)**



### 2.3 Bayesian sea ice probability

To calculate the Bayesian sea ice probability, the algorithm computes the minimum normalized squared distance (or maximum likelihood estimator, MLE) from observations $\sigma_i^0$ to the sea ice $\sigma_{ice}^0$ and ocean wind $\sigma_{ocean}^0$ model functions:

$$MLE_{ocean} = \sum_{i=1,\ldots,N}\left(\sigma_i^0 - \sigma_{ocean,i}^0\right)^2 / var[\sigma_{ocean,i}^0] \qquad (1)$$

$$MLE_{ice} = \sum_{i=1,\ldots,N}\left(\sigma_i^0 - \sigma_{ice,i}^0\right)^2 / var[\sigma_{ice,i}^0] \qquad (2)$$

where N is the number of instrument looks (N=4 for QuikSCAT, N=3 for ERS/ASCAT) and the model variances describe the tolerable (statistical average) range of departures to the GMF:

$$var[\sigma_{ocean}^0] = \left(K_p^2 + K_{geo}^2\right)\sigma_{ocean}^{0\,2} \qquad (3)$$

$$var[\sigma_{ice}^0] = \left(C_{mix}K_p\sigma_{ice}^0\right)^2 \qquad (4)$$

where $K_p$ represents instrumental noise, $K_{geo}$ is a measure of backscatter variability due to wind variability within the sensor footprint, and $C_{mix}$ is a tolerance factor introduced in the sea ice model variance to allow for backscatter variability introduced by mixed open water and sea ice conditions. The conditional open water and sea ice probabilities are represented by chi-square distributions with N-1 and N-2 degrees of freedom for the sea ice and ocean wind classes:

$$p(MLE_{ice}) = \chi_{N-1}^2(MLE_{ice}) \qquad (5)$$

$$p(MLE_{ocean}) = \chi_{N-2}^2(MLE_{ocean}) \qquad (6)$$

And the daily Bayesian sea ice probability is finally calculated as:

$$p(ice|\sigma) = \frac{p(\sigma|ice)p_0(ice)}{p(\sigma|ice)p_0(ice)+p(\sigma|ocean)p_0(ocean)}, \qquad (7)$$

where $p_0(ice)$ and $p_0(ocean)$ are the *a priori* probabilities derived from previous observations, and $p(\sigma|ice)$ and $p(\sigma|ocean)$ are the conditional open water and sea ice probabilities defined in Eqs. (5-6). In principle, we do not grant any
statistical relation between sea ice probability and sea ice concentration.

### 2.4 Normalized sea ice backscatter

While the QuikSCAT mission observes backscatter at Ku-band from an incidence angle of 54 deg (outer VV-pol beam) and 46 deg (inner HH-pol beam), the ERS and ASCAT missions observe backscatter at C-band from a broad range of incidence angles collected across the swath (from 18 to 64 deg in VV-pol). In order to build a uniform record of sea ice backscatter, all
the C-band measurements must be normalized to a standard incidence angle (chosen 52.8 deg, set in the middle of the ASCAT mid-beam swath, and closest to the QuikSCAT VV-pol incidence) using a model that describes the dependence of C-band sea ice backscatter on incidence angle. In the present version, the normalization assumes a linear relation between C-

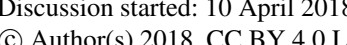



band backscatter and incidence angle using sea ice type dependent coefficients (Ezraty and Cavanie, 1999). A refined incidence angle correction based on the empirical C-band sea ice backscatter model developed in (Otosaka et al., 2017) is planned for a future release. The largest obstacle, though, arises from the presence of composite C and Ku-band observations in a single backscatter record, since their sensitivities to sea ice type differ. Both frequencies are similarly responsive to

5 surface roughness, e.g., over deformed first year sea ice, but Ku-band is more responsive to volume scattering in multiyear ice (Ezraty and Cavanie, 1999). As a result, the separability between deformed FY and MY ice classes is better at Ku-band. At C-band, the disambiguation between deformed FY and MY ice classes is more difficult in terms of backscatter alone, although recourse can always be made to additional information, such as the monitoring of backscatter derivatives, or the introduction of geographical constraints, such as a marginal sea mask.

**3 Historical record**

**3.1 Sea ice extents**

The time series of Arctic and Antarctic sea ice extents observed by the ERS, ASCAT and Quikscat scatterometers from 1992 to 2016 are shown in Fig. 4, along with the differences to the sea ice extents from the SSMI(S) passive microwave sea ice concentration (15% threshold) algorithms from the NSIDC-0051 (Cavalieri et al., 2015) and the OSISAF's latest major

reprocessing release [OSISAF-409a as in (Tonboe et al, 2015) extended into 2016 with OSISAF-450] in Fig. 5. The sea ice extents are constrained by a unique land mask built from the union of all active and passive sensors' land masks. No significant long-term trends are observed in the active-to-passive differences, other than a slight increase in the variability of the Arctic sea ice extent biases from 1992 to 1996, which has been attributed to data gaps in the ERS-1 mission due to SAR operations (Otosaka et al., 2017).

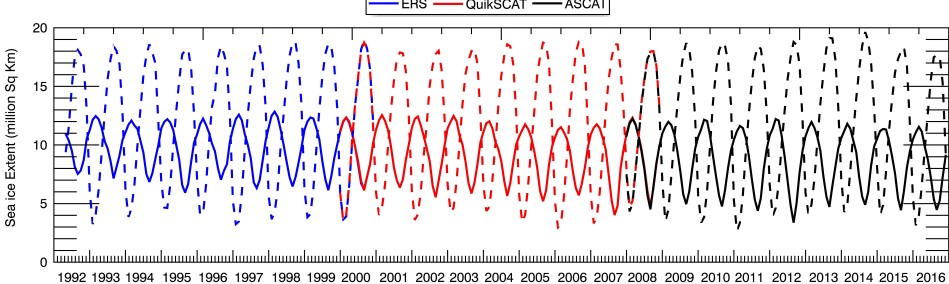

**Figure 4: Time series of monthly Arctic (continuous) and Antarctic (dashed) scatterometer sea ice extents from 1992 to 2016.**





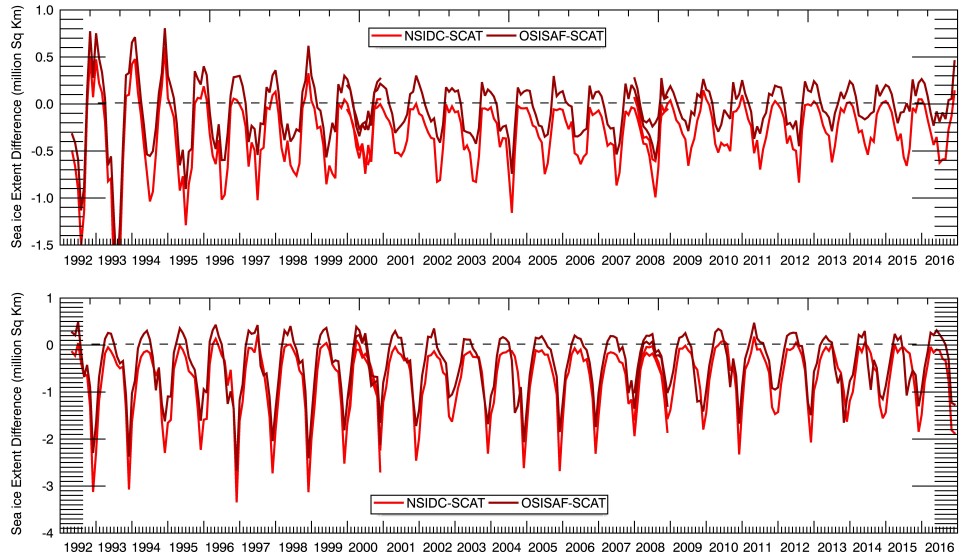

**Figure 5: Time series of scatterometer sea ice extent differences to passive microwave (SSMI-based) products from NSIDC-0051 and OSISAF-409a from 1992 to 2016 for the Arctic (top) and Antarctic (bottom).**

The correspondence between the Quikscat scatterometer and passive microwave sea ice extents from the NSIDC (NT-based) algorithms has been extensively verified using coincident SAR imagery (Belmonte Rivas and Stoffelen, 2011) to reveal excellent agreement during the winter and fall seasons, and persistent differences during the spring and summer months. Figure 6 illustrates the seasonal pattern of active-to-passive sea ice extent biases over the Quikscat-to-ERS and Quikscat-to-

ASCAT transition periods in 2000 and 2008. The agreement between the overlapping C and Ku-band scatterometer sea ice extents is very good all year round, with differences within 0.25 million km$^2$ and an estimated ice edge accuracy of about 20 km. The agreement between the scatterometer and passive microwave sea ice extents is of comparable high quality during the freezing season, but diminishes during the melting (spring and summer) months, noting that the amplitude of summer biases in the OSISAF-409a product is smaller than in the NSIDC-0051 product, particularly for the Arctic sea ice extents.

The largest sea ice extent biases occur at the end of the summer, reaching from 0.6 to 2.0 million km$^2$, and corresponding to estimates of the minimum sea ice extent that may differ by up to 10-30%. As a result, the expression of the Arctic minimum sea ice extent in the scatterometer record may occur up to 15 days later than with passive microwaves. Figure 7 illustrates a typical spatial layout of active-to-passive sea ice extent biases for a particular late summer day (15[th] September 2016). The collocated NIC chart for this particular day, which delineates the subjective extent of the summer sea ice pack (with sea ice

concentrations larger than 80% and marginal sea ice excluded), showcases the higher sensitivity of the scatterometer record



to lower concentration and water-saturated sea ice conditions, particularly over the confluence of the Chukchi and East Siberian Seas, along with the large differences in sea ice concentration estimates from different passive microwave algorithms (of up to 30%) in the central Arctic (Ivanova et al., 2015).

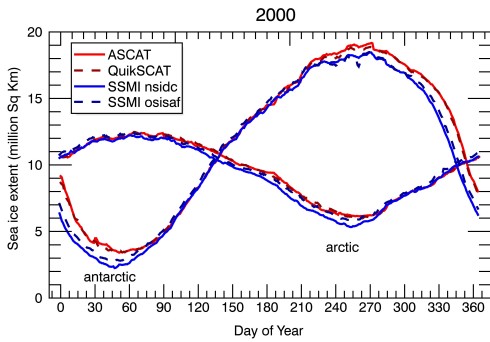 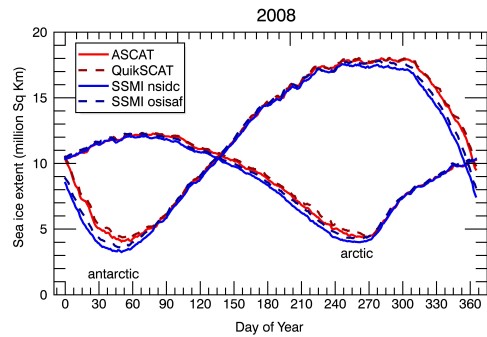

**Figure 6: Comparison of ERS, QuikSCAT, ASCAT and passive microwave (OSISAF-409a and NSIDC-0051) sea ice extents over 2000 (left) and 2008 (right).**

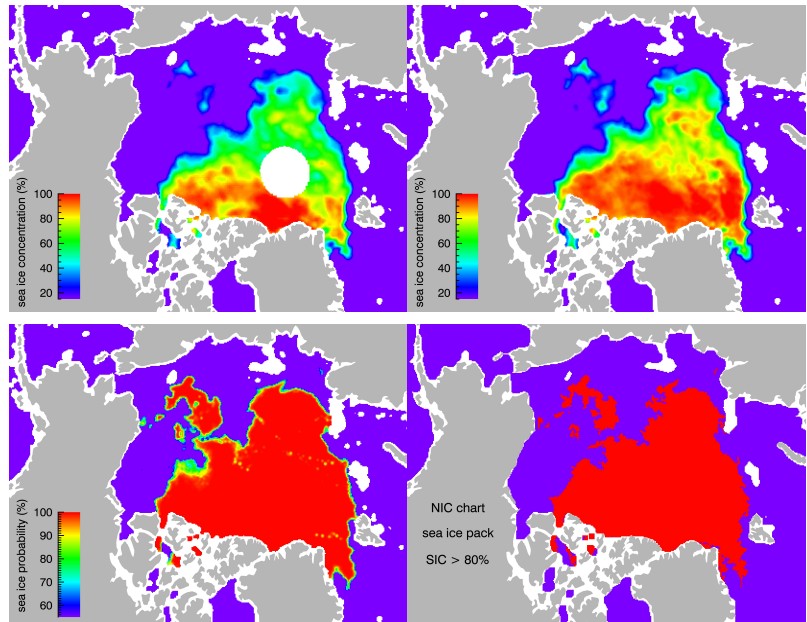

**Figure 7: Comparison of summer sea ice extents on September 15ᵗʰ 2016 from passive microwaves (NSIDC-0051 top left, OSISAF-450 top right), active microwaves (ASCAT, bottom left) and NIC sea ice charts (bottom right). The color scales represent sea ice concentration (top panels), sea ice probabliity (bottom left panel) and NIC sea ice concentration larger than 80%.**



Surface wetness and melt ponding are thought to be responsible for large errors in passive microwave sea ice concentrations during spring and summer [Comiso and Kwok, 1996], and these errors affect the ocean heat contents and associated surface fluxes when assimilated into ocean and atmosphere reanalyses [Hirahara et al., 2016]. In this context, the scatterometer

record nicely complements the passive microwave products in monitoring the occurrence of melt ponding, and delineating the expanse and evolution of the rotten late summer ice classes.

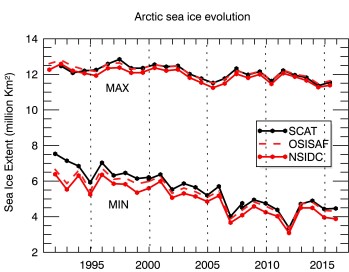
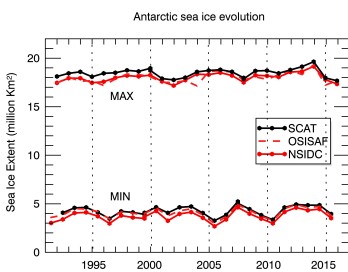

**Figure 8: Evolution of minimum and maximum monthly sea ice extents from scatterometers (black line) and passive microwaves (continuous and dashed red lines) from 1992 to 2016 for the Arctic (left) and Antarctic (right).**

Figure 8 shows the long-term evolution and inter-annual variability of the Arctic and Antarctic minimum and maximum sea ice extents from the scatterometer and passive microwave records. These figures attest to the coincident emergence of significant events, such as the Arctic summer minima in 2007 and 2012, or the Antarctic wintertime maxima in 2014, on top of long-term trends that bear witness to Arctic sea ice decline, and a modest increase in Antarctic sea ice extents. Note that

while the NSIDC algorithm ranked the summer of 2016 as second lowest in Arctic sea ice extent, tied with 2007 (NSIDC, 2016), the scatterometer record observes a somewhat slower trend in the decline of Arctic summer ice over the last 5 years, and only ranks 2016 as fifth lowest [with 4.5 (3.9) million km$^2$ according to the scatterometer (NSIDC) record].

### 3.2 Sea ice backscatter

The monitoring of sea ice backscatter may be used to discriminate Arctic FY and MY sea ice types, but it also can be applied

to estimate sea ice motion by feature tracking (Zhao, Liu and Long, 2002; Lavergne et al., 2010), characterize Antarctic sea ice types (Morris, Jeffries and Li, 1998; Haas, 2001; Willmes, Haas and Nicolaus, 2011) or estimate the onset and duration of melt (Drinkwater and Liu, 2000; Howell et al., 2008). As already noted, the discrimination between Arctic FY and MY ice types using active microwaves is not without difficulty, its main hindrances being the seasonal variability of backscatter, including the effects of surface deformation, ice/snow metamorphism and a developing snow cover, or the arrival of summer

signatures, more dependent on surface weather via processes such as wet snow attenuation and changes in brine temperature



(Barber and Thomas, 1998). The annual cycles of MY ice coverage in the Arctic Ocean have been estimated using the QuikSCAT record (1999-2009) by (Kwok et al., 2009) using a fixed backscatter threshold from January to April, and by (Swan and Long, 2012) using a seasonally dependent backscatter threshold from November to April, to produce a multi-mission record extended forward in time onto 2014 using Ku-band Oceansat-2 scatterometer (OSCAT) data (Lindell and

Long, 2016). In order to avoid the high-backscatter FY ice in the marginal ice zone (MIZ) from being classified as MY ice, (Kwok et al., 2009) introduced a static geographical mask, while (Lindell and Long, 2016) applied a MIZ correction algorithm based on the temporal persistence of the MY signature, along with a 40% sea ice concentration mask from passive microwave data.

For the determination of the time series of Arctic MY ice coverage, we adopt the single backscatter threshold approach. To avoid dealing with seasonal variability, we only use stable wintertime (March) backscatter maps, assuming that the backscatter signatures of the reference winter sea ice classes do not change with time. We also introduce a geographical mask to screen the high backscatter response from MIZ sea ice, which has been attributed to surface deformation by compression and irreversible snow/ice metamorphism after melt-freeze events (Voss et al., 2003) (Willmes et al., 2011). The

geographical mask delimits the Arctic Basin (see red contours in Fig. 11) across the Fram Strait and Svalbard, to Severnaya Zemlya through Franz Josef Land (Kwok, Cunningham and Yueh, 1999). An additional line from Point Barrow to Wrangel Island also excludes the Chukchi Sea from the MY area estimations. The geographical mask omits the ubiquitous presence of multiyear ice in the Greenland Sea, or its episodic incursions into the marginal Chukchi, Barents and Kara Seas.

For the consistency of the record, the backscatter thresholds for MY ice detection at Ku and C-band are matched attending to their joint backscatter distributions and resulting spatial boundaries. The top panels in Figure 9 show the joint backscatter distributions of Arctic sea ice at C-band and Ku-band for the month of March in 2000 and 2008, before application of the geographical mask. Before masking, the joint distributions of wintertime sea ice backscatter are characterized by two elongated clusters: an upper cluster corresponding to perennial (MY) ice, and a lower one corresponding to seasonal (FY) ice

(Ezraty and Cavanie, 1999). The cluster elongation gives account of geophysical variability, with perennial ice types getting brighter as they accumulate summer conditions, and seasonal ice types becoming brighter with surface deformation and/or metamorphism. Note that the range of backscatter variability associated to deformation and/or metamorphism in the lower seasonal ice cluster (~ 5 dBs) is comparable at C-band and Ku-band. The signature of volume scattering, though, is stronger at Ku-band, and effective at separating the rough FY and MY ice domains, which remain partly overlapping at C-band. The

bottom panels in Figure 9 illustrate the effectiveness of the geographical mask at removing the MIZ signature, and how necessary this is for the definition of an effective separation threshold between FY and MY classes at C-band. Starting from the already established Ku-band threshold of -14.5 dB for the Quikscat VV backscatter, which would correspond to a MY sea ice fraction of 30% according to RADARSAT (Kwok, 2004), and aided by the correlation of the MY ice spatial




boundaries at Ku and C-band (see Figure 11), an optimal threshold for MY detection using C-band VV backscatter (52.8 deg incidence) is found at -18.3 dB.

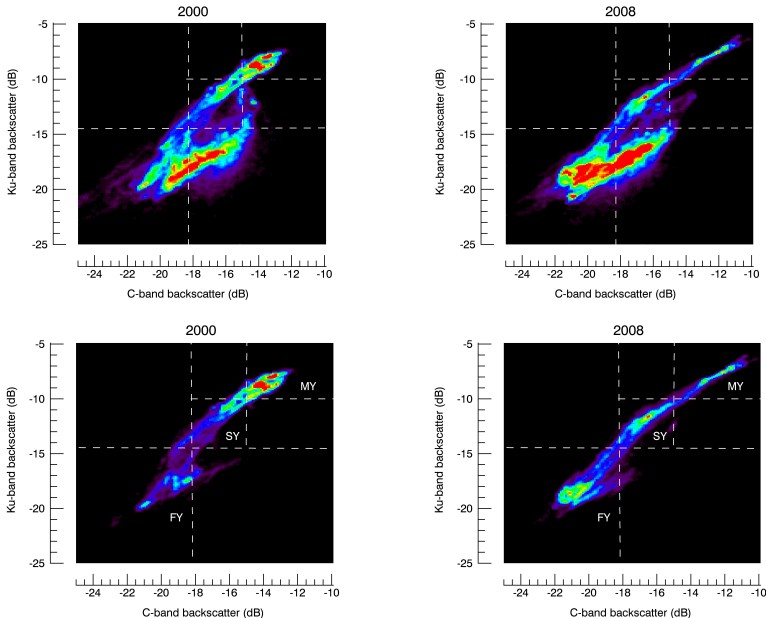

**Figure 9: Joint distributions of wintertime (March) sea ice backscatter at C-band (x-axis) and Ku-band (y-axis), before (top row) and after (bottom row) applying the geographical mask in 2000 (left, ERS vs QSCAT) and 2008 (right, ASCAT vs QSCAT).**

The left panels in Figure 10 show the marginal distributions of sea ice backscatter at Ku-band (top) and C-band (bottom) that
10  correspond to the geographically masked joint distributions for the year 2000. The marginal backscatter distributions are characterized by two well-defined modes, associated to FY and MY sea ice types, connected by a transition range. Using spatially collocated Ku and C-band backscatter measurements, sea ice types in the transition range between the FY and MY modes may be further separated (see joint distribution in the bottom left panel of Fig. 9) into deformed FY (in the high range of the seasonal sea ice cluster), SY ice (in the low range of the perennial sea ice cluster), and FY-MY mixtures (along the
15  path connecting the seasonal and perennial clusters). We note that an optimal threshold for MY detection should guarantee that most of the rough FY is removed from the MY category, while collecting various fractions of SY, MY, and FY-SY-MY mixtures.





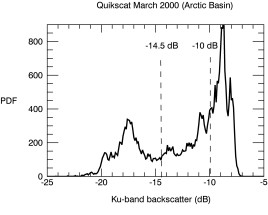 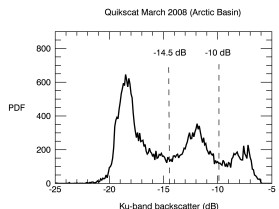

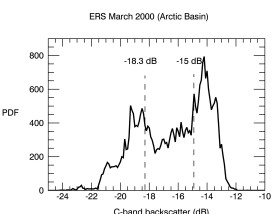 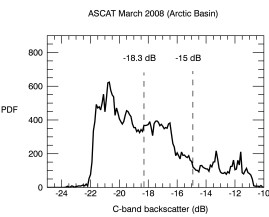

**Figure 10: Marginal distributions of wintertime (March) sea ice backscatter collected at Ku-band (top row) and C-band (bottom row) after applying the geographical mask in 2000 (left column) and 2008 (right column).**

After the anomalously large loss of Arctic sea ice that occurred in the summer of 2007, the shape of the wintertime sea ice backscatter histograms have become remarkable altered. The earlier bimodal (FY+MY) histograms have been replaced by trimodal distributions, featuring a smaller MY mode, and a new mode corresponding to SY ice emerging in the low range of the perennial sea ice cluster (see bottom right panel in Fig. 9). The emergence of the new SY mode is also evident in the marginal distribution of Ku-band backscatter for the year 2008 (see top right panel in Fig. 10), though more difficult to see in

the marginal distribution of C-band data for the same year (see bottom right panel in Fig. 10, around -16.5 dB) because of the larger influence of deformed FY in this frequency and backscatter range. In order to monitor the evolution of the newly emerged SY mode, we split the perennial ice cluster into separate SY and old MY classes using an additional set of thresholds (-10 dB for Ku-band and -15 dB for C-band) whose location relative to the original FY and MY modes is shown in the joint and marginal distributions in Figs. 9 and 10.

The spatial distributions of the FY, SY and old MY classes that result from applying the single threshold approach on Ku and C-band backscatter images are displayed in Figure 11, along with the average sea ice age from the EASE-Grid dataset NSIDC-0611 from (Tschudi et al, 2016) for that period. The spatial distributions of the total MY ice class (defined as the sum of SY and old MY classes) from the scatterometer and the lagrangian sea ice age analyses are in general good agreement, although their depictions of the SY ice class differ somewhat. We note that the old MY sea ice class has a larger

geographical spread in the lagrangian dataset, particularly over areas where MY ice is exposed to strong shear stress, such as in the Beaufort Sea. From the analysis of joint backscatter distributions, we know that the scatterometer SY class is bound to contain varying amounts of FY-SY-MY mixtures (and probably some deformed FY too). However, the lagrangian dataset is





monitoring the age of the oldest ice in a cell, regardless of its weight over other ice fractions, probably biasing this product towards a larger spread of old MY ice in the Arctic.

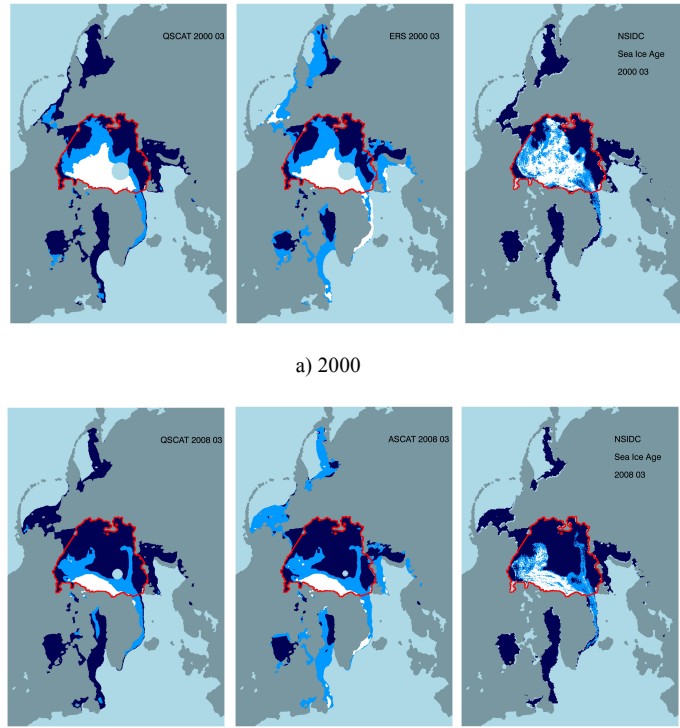

a) 2000

a) 2008

**Figure 11: Geographical boundaries of wintertime FY (dark blue), SY (light blue) and MY (white) sea ice classes from Ku-band backscatter (left panel), C-band backscatter (middle panel) and NSIDC sea ice age (right panel) for 2000 (top row) and 2008 (bottom row). The contour of the geographical mask used to delimit the Arctic Basin is shown in red.**

The time series of the total MY sea ice extents, along with the extents of the separate SY and old MY class contributions calculated using the backscatter threshold approach on wintertime (March) data collected within the geographically masked Arctic Basin is shown in Fig. 12. All estimates exclude a common polar gap extent of 0.354 million km$^2$ around the North Pole. The evolution of the total MY sea ice extents derived from the scatterometer record agrees well with that derived from the NSIDC sea ice age dataset, showing a MY pack that begins to lose balance around 2005, after several consecutive years of decline, to finally collapse into a large loss in 2007. The partition into total SY and old MY ice extents is also similarly depicted in both datasets, regardless of discrepancies in their spatial distributions, providing further evidence to support our claim of a newly emergent SY ice mode. Figure 12 proves that the largest decline in Arctic MY ice is borne by loss of old

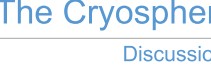
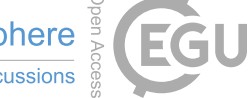

MY ice after 2007, with a more steady production of SY ice partly buffering those losses, and driving later recovery events
such as observed in 2014.

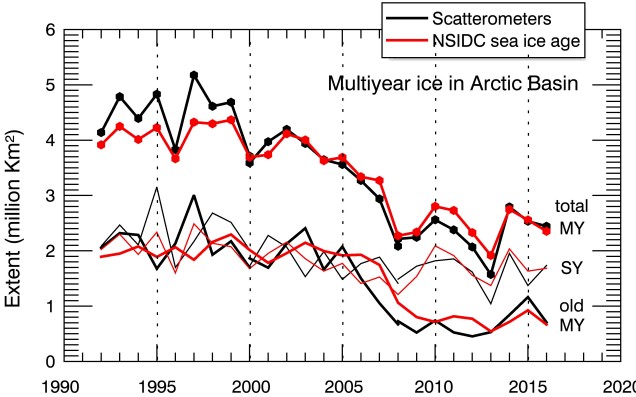

**Figure 12: Time series of monthly wintertime (March) total multiyear sea ice extents (segmented into SY and older MY classes)**
**within the Arctic Basin from the scatterometer (black) and the NSIDC sea ice age (red) records.**

Thus far, we have justified the definition of a separate SY ice class after the emergence of a SY mode with an entity of its
own in the scatterometer backscatter histograms. The differentiation of SY and lower concentration of MY using a single
frequency remains an open question though. By construction, the scatterometer SY class will accommodate various fractions
of deformed FY and FY-SY-MY mixtures in it, which we suggest may be differentiated from the homogenous SY ice
signature by recourse to dual Ku-band and C-band observations. In this context, the reprocessed Ku-band Oceansat-2 record
spanning the period from 2009 to 2014, also available in our scatterometer record, affords new opportunity to resolve this
ambiguity. We note that the scatterometer record may be helpful as a check against currently developing algorithms for MY
ice concentration based on satellite passive microwave or blended data, given that none of these latter products uses a
separate tie point for SY ice, leaving the SY ice signature to be effectively interpreted as lower concentration MY ice. As an
illustration, Figure 13 shows the spatial distribution of MY ice according to a selection of state-of-the-art products for the
month of March 2016, including sea ice age [from the scatterometer record, the NSIDC record of (Tschudi et al, 2016), and
the SICCI record (Korosov et al, 2017)], MY ice concentration [from the OSISAF-403 (Aaboe et al, 2016), the U. Bremen
algorithm (Ye et al, 2016), and the SICCI algorithm (Korosov et al, 2017)], and sea ice thickness from the AWI Cryosat-2
dataset (Ricker et al, 2014)].

Even though the general representation of MY ice is similar across all products, there are remarkable differences as well,
mainly regarding the distribution of the old MY ice class north of the Canadian Arctic Archipelago (CAA, with large
variations across the sea ice age records), and the presence of MY ice north of the Beaufort and Chukchi Seas (with notable




differences between the MY ice concentration records). The ice thickness product is revealing in that the thickest sea ice (more than 3 m thick, and most likely associated to old MY ice) appears mostly confined to a thin strip along the CAA shore (in agreement with the scatterometer old MY ice class), and that it shows no traces of thick ice north of the Beaufort and Chukchi Seas (in disagreement with some of the MY concentrations, and the NSIDC sea ice age record). Further, we note a

5 large extension of very thick ice (more than 3 m thick) north of Greenland, which is labelled as SY ice in the scatterometer record (probably ridged SY ice converging into Fram Strait), which effectively appears as low concentration MY ice in the Univ. Bremen and SICCI algorithms, suggesting problems with the tie points definition in MY ice concentration algorithms (not in the OSISAF-403 dataset, because it is reports MY presence, not concentration). Finally, we find relative good agreement between the scatterometer SY ice class and the 2.0 m isoline from the ice thickness record, suggesting the

10 utilization of the backscatter record as a reliable proxy for the estimation of thick sea ice thickness in the Arctic, taking into account its limitations (e.g. spurious high backscatter north of Franz-Josef Land).

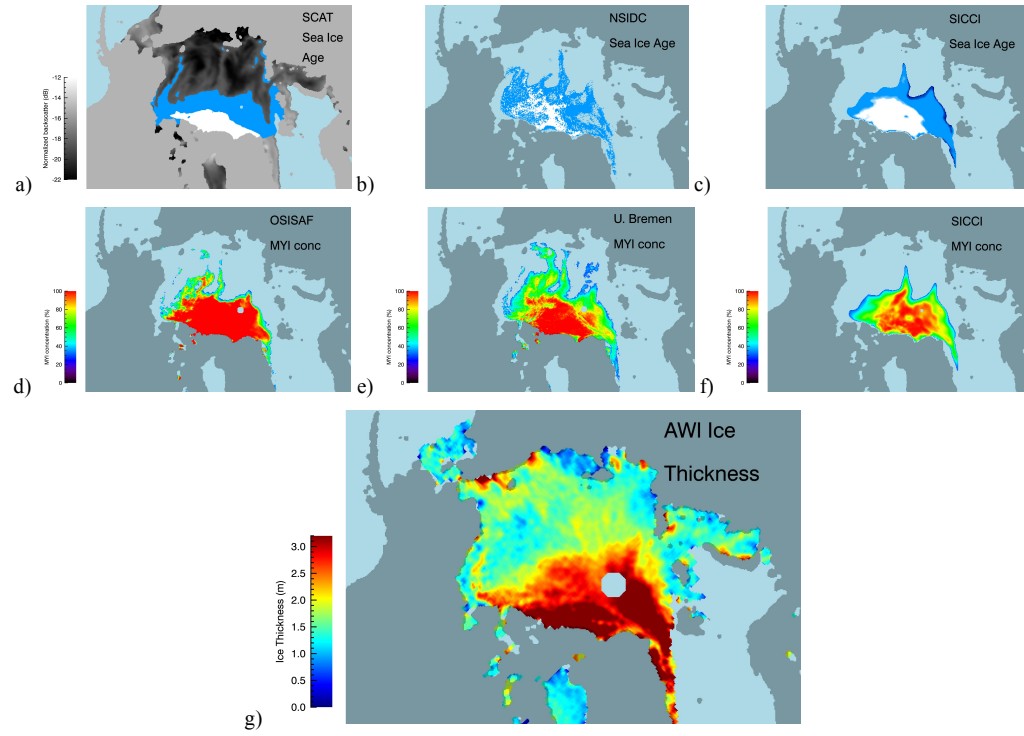

**Figure 13: Different observation-based products for the representation of multiyear ice in March 2016: by sea ice age (top row, light blue is SY ice, white is old MY ice) from the scatterometer record (left), the NSIDC sea ice age (middle) and the SICCI algorithm (right); by multiyear ice concentration (middle row, MY>30%) from the OSISAF-403 (left), the Univ. Bremen algorithm (middle) and the SICCI algorithm (right); and by sea ice thickness (bottom plot).**





Noting the lack of extensive in-situ validation sources for satellite-based datasets, one should rely on consistency among products as the best approach to check retrievals. Yet, the differences just noted in this section make it clear that further effort is necessary towards the optimal integration of active and passive microwaves, not only for the classification of sea ice
types, but for the determination of summer sea ice edge and concentrations.

## 4 Conclusions

We present the first inter-calibrated long-term record of sea ice extents and backscatter derived from satellite scatterometer missions (ERS, QuikSCAT and ASCAT) extending from 1992 to present date. The scatterometer record, whose continuation into the future is guaranteed by the Metop ASCAT (B and C) and EPS-SG series, provides a valuable independent account
of the state of Arctic and Antarctic sea ice cover, with daily sea ice extent and backscatter maps available at www.knmi.nl/scatterometer/ice_extents.

The scatterometer sea ice extents show excellent agreement with passive microwave fields in the fall and winter seasons, with differences within 0.25 million km$^2$ and an estimated ice edge accuracy of about 20 km, but show enhanced sensitivity
to lower concentration and water-saturated sea ice conditions during the spring and summer months, as verified by comparison to NIC sea ice charts. The sea ice concentrations derived from satellite passive microwave brightness temperatures are affected by surface wetness during the melt season, typically underestimating the summer sea ice concentration and summer sea ice extent by up to 30%, and having a non-negligible impact on the ocean heat contents and surface fluxes when assimilated into reanalyses. In this context, the scatterometer sea ice extents and probabilities nicely
complement the passive microwave products in providing a solid basis to monitor the occurrence of sea ice concentration errors due to melt ponding, and to delineate the expanse and evolution of the rotten late summer ice classes.

The scatterometer backscatter maps also provide enhanced means to differentiate between sea ice types. Our study of the evolution of the wintertime seasonal (FY) and perennial (MY) ice classes in the Arctic Basin from 1992 to present day
shows, in good agreement with the NSIDC sea ice age dataset, a MY ice pack that begins to lose balance around 2005, after several consecutive years of decline, to finally collapse into the a record loss in 2007. The scatterometer maps also reveal the emergence of a new mode in the backscatter histograms after the record sea ice loss in 2007, bearing striking resemblance in both temporal evolution and spatial distribution with the SY ice class of the NSIDC sea ice age dataset. Monitoring the evolution of this newly emerged SY class reveals that the largest decline in Arctic MYI ice is borne by loss of old MY ice,
with a more steady production of SY ice partly buffering those losses, and driving later recovery events such as observed in 2014.

We note that the differentiation between SY, deformed FY and lower concentration (but older) MY ice may be difficult using backscatter from single frequency. However, the simultaneous combination of C-band and Ku-band backscatter measurements allows further differentiation of sea ice types into deformed FY (high C-band, low Ku band), SY ice (low C-band, high Ku-band) and FY-MY mixtures (moderate Ku and C-band responses), suggesting a new approach to their disambiguation. As such, the availability of coincident Ku-band and C-band missions (such as during the Quikscat overlap in 2000 and 2008, or the Oceansat-2 overlap from 2009 to 2014) affords new promise.

The scatterometer backscatter record is helpful as a check against currently developing algorithms for MY ice concentration based on satellite passive microwave or blended data, given that none of these latter products uses a separate tie point for SY ice, leaving the SY ice signature to be effectively interpreted as lower concentration MY ice. The comparison of a selection of state-of-the-art datasets for the representation of MY ice (including sea ice age, MY ice concentration and ice thickness estimates) in the Arctic reveals some notable inconsistencies, mainly regarding the ambiguity between compact SY and lower MY ice fractions, the spatial distribution of old MY ice in the sea ice age records, and the apparently spurious presence of MY ice in the Central Arctic in some of the MY concentration records derived from satellite passive microwaves. The relative good agreement between the backscatter-based sea ice (FY, SY and older MY) classes and the ice thickness record from Cryosat suggests its applicability as a reliable proxy in the historical reconstruction of sea ice thickness in the Arctic.

Noting the lack of extensive in-situ validation sources for satellite-based datasets, one should rely on consistency among products as the best approach to check retrievals. Yet, the differences among state-of-the-art products noted in this paper make it clear that further effort is necessary towards the optimal integration of active and passive microwaves, not only for the classification of sea ice types, but for the determination of summer sea ice edge and concentrations.

**Acknowledgments**

The authors would like to acknowledge the financial support of the ESA Scirocco project, the EUMETSAT OSI-SAF for the provision of backscatter data, our KNMI colleague Jeroen Verspeek for his insights, and the NASA and NSIDC public data archives as essential towards the completion of this activity. Processing of the AWI CryoSat-2 (PARAMETER) ice thickness is funded by the German Ministry of Economics Affairs and Energy (grant: 50EE1008) and data obtained from http://www.meereisportal.de (grant: REKLIM-2013-04).

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
