# Peer review of "A scatterometer record of sea ice extents and backscatter: 1992-2016"

_The Cryosphere, 2018_

## Short Comment (SC1) · 11 Apr 2018

Your conclusions (pg 18 para 5) correctly refer to the potential of combinations of contemporaneous scatterometer data at Ku-band and C-band, together with multifrequency passive microwave data. However, arguably this is not new promising potential. Max. Likelihood, Baysian classifications of sea ice using combined active/passive microwave were performed and published already in Remund et al (2000) An Iterative Approach to Multisensor Sea Ice Classification, IEEE TRANSACTIONS ON GEO-SCIENCE AND REMOTE SENSING, VOL. 38, NO. 4, JULY 2000. It would be helpful to refer to this work.

Arguably the task at hand is simply to go ahead and to use all fundamental climate

data records from Scatt (SASS, ERS AMI, NSCAT, QSCAT, MetOp, OSCAT, etc.) with Passive Microwave (ESMR, SMMR, SSM/I, AMSR, MWRI, etc.) to generate a combined timeseries of areal estimates of different ice classes which can be used to guide reanalyses. They can also be combined with the contemporaneous altimetry ice thickness datasets to derive the true Essential Climate Variable of interest - sea ice volume, broken down into the different ice classes.

With Antarctification of the Arctic, the ice pack will become more seasonal, and multiyear ice progressively replaced, and thus the Antarctic example already published is quite relevant in this context. Mark Drinkwater
Fig. 5. RGB composite image of the first three principal components for 1996 JD 261-266. The red channel is the top principal component image, the green is second, and blue is third. The image is useful in evaluating the type of information contained in the top three PCA scores. The six training regions are also indicated.

Fig. 1.

---

## Short Comment (SC2) · 16 Apr 2018

Thank you Dr. Drinkwater for your useful comment. We will add this reference in our revised manuscript. We agree that the optimal combination of all the data records available will be most beneficial to the creation of a long-term dataset of sea ice class extents and volume estimates

---

## Referee Comment (RC1) · Anonymous Referee #1 · 5 May 2018

The authors construct a data record of sea ice extents and ice type coverage (multiyear, second year, and first-year ice) from satellite scatterometers (ERS, QuikSCAT, ASCAT) that show the loss of sea ice and old ice over the last 25 years. Relatively good agreements between thicknesses classes from CryoSat-2 and the ice types suggest that the ice types could be reliable proxies of sea ice thickness in the Arctic.

While the approaches to derivation of the records (ice extent and ice type) are reasonable, the analysis of data quality, and the conclusions (and therefore the abstract) are rather qualitative and require some tightening up. If these data sets were to be presented as climate quality, then I should expect a more detailed assessment of the data quality and consistency (i.e., quantify the differences between the different time series and their trends). In particular:

[Figure]

1. While I understand it is difficult to address the absolute uncertainties of these retrievals, I believe that the authors should at least address the potential variability in the estimates associated with the calibration of the scatterometers, especially since fixed thresholds are used in the separation of the ice classes.

2. Could the backscatter signature of SY ice just be a mixture of MY and FY-ice at the regional transition between the regions with the two dominant ice types? This should be addressed.

3. The authors' statement that: '...we find relative good agreement between the scatterometer SY ice class and the 2.0 m isoline from the ice thickness record, suggesting the utilization of the backscatter record as a reliable proxy for the estimation of thick sea ice thickness in the Arctic...' is not really supported by the analysis provided here. To demonstrate that the backscatter record is a proxy of thickness would entail more work. For example, as an assessment, the authors could examine the mean ice thickness of the three ice types over the CryoSat-2 period and then examine the variability within the different categories. In any case, if the authors were to clarify what they meant by 'proxy' it would be more satisfactory.

I think these issues should be addressed.

---

## Referee Comment (RC2) · Anonymous Referee #2 · 12 Jun 2018

General comments:

The manuscript "A scatterometer record of sea ice extents and backscatter: 1992–2016" by Maria Belmonte Rivas et al. (tc-2018-68) introduces a data record of sea ice extent and backscatter merging different space-borne scatterometers (ERS, QuikSCAT, and ASCAT). In addition to presenting the methodology and resulting data record, the authors use the new almost 25 years long data to describe recent changes of Arctic sea ice observed after the September 2007 minimum.

The contribution is generally of fair quality, and the text is understandable. The paper could be improved by giving some more details on the methodology, and discuss uncertainty of the retrievals.

[Figure]

A major concern are the shortcuts taken when comparing the capabilities of scatterometers (this study) versus (passive microwave) radiometers (e.g. NSIDC and OSISAF SIC data records), especially when it comes to summer melt. The claims of the authors on that specific topic are too often not supported by facts.

I encourage the authors to revise the way their findings are presented (not the methodology they apply to compute the data records) and to consider the comments below as an incentive to enhance the quality of their contribution. They should strive at being more balanced, and avoid shortcomings in the presentation of their results.

As a general comment : all the plots (especially the maps) are too small.

Specific comments:

Abstract: "providing a means to correct for summer melt ponding errors". This is an overstatement. Unless you include in your study an investigation of the melt-pond season (May-June-July), for example using independent melt-pond fraction information (as done in Kern et al. 2017), the statement is not supported. You should be cautious with any statements about the alleged superiority of scatterometers to measure sea ice under melt-ponding conditions : the radar backscatter signal will be very much influenced by the surface water, and by the melting snow. In these conditions, it is hard to believe that scatterometers will have the accuracy to partition the 0%-30% reduced ice concentration (as measured by passive microwave) into some percents of pure "open water" contribution and some percents of pure "melt water" contribution. Such a partition however is what it would take to "provide a means to correct for summer melt ponding errors".

page 1 line 29: "instances of missing thin ice" . . . from where is it missing? from the passive microwave records?

page 2 line 5: when comparing sea ice estimates to operational ice charts, especially during summer, one should always question the accuracy of the charts.

[Figure]

page 2 line 14 : "0.1 dB via buoy collocation" (do you mean "via collocation of wind retrievals at ocean buoy locations"?)

page 2 line 16 and 17: "which are known to cause discontinuities...". please consider the following wording: "which -if not done properly- can affect long-term trends in sea ice concentration". It is fortunate that the inter-calibration of brightness temperatures is generally well taken care of by expert teams prior to computing data records of sea ice concentration. Titchner and Rayner (2014) discuss the stitching of SIC from different sources (passive microwave, navigational ice charts, etc..) and not the impact of non-optimal inter-calibration of passive microwave sensors can have on SIC trends.

page 2 offers a good occasion to re-state that passive microwave records are primarily aiming at mapping sea ice concentration, and sea ice extent is not a goal as such. sea ice extent is only a downstream indicator.

page 2 line 21: "It is known that...". Then please rather add a good citation.

page 3 line 4: when referring to the ice age data from NSIDC, a more recent citation is 10.1109/JSTARS.2010.2048305.

page 3, line 20: which Metop platform(s) are used for ASCAT? please specify in the text.

page 3 line 25: It would be useful to mention here the zenith angles of the different instrument (for example include here the first sentence of section 2.4).

page 3 line 25 : is it worth briefly mentioning NSCAT and why it was not selected in your data record?

page 5, line 5: "stable wintertime backscatter levels" I understand this as your ice GMF is static, and not dynamically updated (e.g. with months). If I misunderstood, please consider making the sentence more explicit.

page 6, line 13: please use one sentence to explain the difference in degrees of freedom.

page 6, eq (7) : this is presented as the formula to compute daily estimates of the probabilities, thus supposedly gathering several passess of a given instrument. Yet, equation 7 does not show any indices. Could you make the indices appear in eq 7? Are the individual probabilities multiplied or averaged together?

page 6: it would be very interesting for the users to extend section 2.3 with some material to describe what would be your approach to providing quantitative uncertainties. A key discussion point might be how to treat that the PDFs of individual observations might be correlated with each others. To the least, uncertainties in the retrievals and how to quantify them should be discussed at the end of the manuscript.

page 6: two informations are missing from section 2: 1) how do you define your SIE (from daily maps of sea ice probability... what is the threshold on probability?) and 2) describe in what sense "these algorithms have been tuned to match the passive microwave sea ice extents during the fall and winter months" (ref your page 2, line 11).

page 8 lines 9, 14, 15, 18: the use of "biases" is problematic as it implies you consider one source (the scatterometer?) is correct and the other (the passive microwave) is at an offset. Replace with "differences" at all 4 occasions.

page 8, lines 7-8 "the agreement... is of comparable high quality during the freezing season"... is it surprising? "these algorithms have been tuned to match the passive microwave sea ice extents during the fall and winter months" (ref your page 2, line 11).

Figure 7: are the colorbars for SIC correct? They would indicate that dark blue is for everything below 20%? It would be interesting to show contour lines of the sea ice extent from NSIDC-0051, OSI-430, and SCATT (e.g. on top of the NIC chart), as this is impossible to observe from the colored maps of SICs.

Concerning Figure 7 : being an operational product meant for tactical navigation in the ice, the NIC ice charts are not a reference for accurate SIC monitoring. It uses

active microwave data (SAR) that might suffer from exactly the same noise sources than the scatterometers, namely that scattered, faceted pieces of ice will seem brighter (higher backscatter) than if the corresponding area of ocean was covered by contiguous ice. Always question the accuracy of NIC charts (or other navigational ice charts), especially during summer.

Page 10: "surface wetness and melt ponding ... during spring and summer [Comiso,...]" 1) consider also citing Kern et al. (2017) https://doi.org/10.5194/tc-10-2217-2016 as a more recent and quantitative assessment of the impact of melt-ponds on the passive microwave retrievals of SIC.

Concerning Figure 6: it shows a mismatch between PMR and SCATT SIE for both hemispheres and the whole of spring+summer seasons, all the way until the SIE minimum. The sentences following Figure 6 (and Figure 7) point at "surface wetness and melt ponding" as a reason for the mismatch. This is not very intuitive, since:

1) the PMR SIE is defined as the area where SIC is larger than 15%. Although the impact of melt-ponding can be up to 20%-30% (Kern et al. 2017) at the maximum of the season, most of the melt-pond covered area will still show SIC>15%, thus only marginally influencing the PMR SIE.

2) Regardless of item 1), melt-ponding (on sea ice) is happening mostly in (late)May, June, and July, (early August) before they drain through the ice. There are no melt-ponds on top of sea ice in mid September (Figure 7). In addition, melt-ponds on top of sea ice are mostly (if not only) observed in the Arctic. Melt-ponds can thus not explain the SH mismatch you document.

However:

1) it is noted that the SCATT SIE has a finer spatial resolution than the SSM/I and SSMIS PMR SIE (this is obvious on Figure 7) due to the size of the FoVs of SSMIS at 19 and 37 GHz. Spatial resolution has been documented to have a definite influence on

the SIE metric (Notz, 2014, https://www.the-cryosphere.net/8/229/2014/). Can spatial resolution have an influence on the SCATT SIE?

2) In the marginal ice zone, geometric scattering effects by disjoint, faceted ice floes will induce higher backscatter, and thus artificially enhanced radar "brightness" than if the same quantity of sea ice was present in a planar, contiguous way. The PMR signal is insensitive to these geometric effects. Could this geometric effect explain the difference in SIE you observe on Figure 6?

3) Finally, if indeed your sea ice GMF is static and tuned for winter conditions (see page 5, line 5), then you should discuss if the spring+summer sea ice GMF is similar (than the winter one) so that to ensure that seasonality of the sea ice GMF does not artificially contribute to seasonality in SCATT SIE.

All in all, and as noted in the introduction to this review : the readability and impact of this paper would be greatly improved if 1) you described the observed differences between PMR and SCATT SIE without too hastily attributing them to deficiencies of the PMR SIEs (specifically melt-ponding), 2) you investigated (or pointed to earlier investigations) how other factors might explain (better?) the differences.

Figure 11: a legend/colorbar is missing. In addition the maps are too small (this applies to all figures in the draft manuscript). Figure 11 shows large discrepancies between the left and middle panels outside the central arctic ocean (e.g. second year ice in Bering Sea in 2008 for ASCAT but not QSCAT). Please either zoom your maps to the Arctic Basin (this is the area you discuss anyway, or comment the large discrepancies (how they can be mitigated).

Page 15, line 18: as noted later in the text, the OSI-SAF product OSI-403 does not hold a MY ice concentration, but a MY/FY classification.

Figure 13: Several remarks:

1) the panels are too small.

2) what date is this from (inside March 2016)? Or is this a monthly average for all panels? If a monthly average you should probably describe how the average is performed, given the variety of input variables (MYI conc, MYI classification, max SIA, mean SIA,...). It would make more sense to have a specific date.

3) consider adding coast lines.

4) legend : the OSI-403 is not a MYI concentration product but a sea ice classification.

5) please add a "first year ice" color (or a sea ice edge line) on panels b) to f).

6) please check your plot c) and f) (from Korosov et al. 2017 data). if this is plotted from variable "sia" (the mean sea ice age), one has to look at 1<sia<2 for the second year ice (as soon as sia is > 1, then it has 2nd year ice contribution). The result will look much more similar to NSIDC SIA and the other maps, especially in the Beaufort Sea.

7) the description of the location of the features is difficult to follow in the text, would it be an idea to write letters on the map (A, B, C, ...) and refer to them?

8) all estimates (but yours in a) show a thin tongue of older ice extending across the Arctic Ocean towards the New Siberian Islands. It can even be noticed somewhat in the Cryosat-2 thickness. It is visible on the shades of "normalized backscatter" in your panel a). Given the variety of methods used for all other panels, are you confident that your new estimate in a) is correct and that there is no such tongue of older ice? Discuss.

Page 17, line 15 : "as verified by comparison to NIC sea ice charts". NIC sea ice charts are not the truth. Please use a different wording than "verify"..

Page 17, line 18 : "typically underestimating the summer sea ice concentration and summer sea ice extent by up to 30%". Which one? SIC or SIE? As discussed above an underestimation of high SICs, even by 30% will have limited impact on the SIE (defined as SIC>15%). Rewrite.

Interactive
comment

page 17, line 21: "providing a solid basis to monitor the occurrence of sea ice concentration errors due to melt ponding". NO. See notes from the introduction. Rewrite.

page 17, line 29, 30: this sentence is an exact copy-paste from previous section. Avoid word-by-word repeats, please.

page 18, line 15 : to use Sea Ice Type (or Age) as a proxy for Sea Ice Thickness is not a new idea. It would be good to cite earlier attempts e.g. Tschudi et al. 2016, http://www.mdpi.com/2072-4292/8/6/457.

Technical corrections:

page 1 line 14: "record sea ice loss" : consider another term than "record" as it is used elsewhere in the paper (including the abstract) as "data record".

page 1 line 19: "1978" (with the start of SMMR ni Oct 1978). There are passive microwave instruments before (e.g. ESMR), but the routine monitoring indeed starts in 1978 with SMMR.

page 3 line 22: "mission transition periods".. consider "mission overlap periods"

page 3, line 13; "measurements about extended" . . . do you mean "around"?

Figure 4: please add a second legend with solid line for NH and dashed line for SH.

Figure 5: please add text in the plot area for NH (top) and SH (bottom).

page 7, line 15 : you are probably referring to OSI-430, that extends OSI-409a (http://osisaf.met.no/p/ice/index.html#conc-reproc). Same in caption to Figure 7.

page 8, line 9 : "Quikscat-to-ERS"... you probably mean "ERS-Quikscat"?

Figure 7: you seem to have a dip in QuikSCAT SIE in antarctic curve for 2000 (around day 260).

page 10, line 24: "arrival of summer signatures" consider "appearance of summer signatures"?

---

## Author Comment (AC1) · 17 Jul 2018

Our replies to referees' comments are inserted in red below.

**Reviewer 1**

The authors construct a data record of sea ice extents and ice type coverage (multiyear, second year, and first-year ice) from satellite scatterometers (ERS, QuikSCAT, ASCAT) that show the loss of sea ice and old ice over the last 25 years. Relatively good agreements between thicknesses classes from CryoSat-2 and the ice types suggest that the ice types could be reliable proxies of sea ice thickness in the Arctic.
While the approaches to derivation of the records (ice extent and ice type) are reasonable, the analysis of data quality, and the conclusions (and therefore the abstract) are rather qualitative and require some tightening up. If these data sets were to be presented as climate quality, then I should expect a more detailed assessment of the data quality and consistency (i.e., quantify the differences between the different time series and their trends). In particular:

Thanks for your review. We admit that the analysis and conclusions are rather qualitative – yet we hope they remain attractive. The purpose of the manuscript is not so much to make any extraordinary statements, but to consider the potential of the scatterometer record to address currently open research questions.

1. While I understand it is difficult to address the absolute uncertainties of these retrievals, I believe that the authors should at least address the potential variability in the estimates associated with the calibration of the scatterometers, especially since fixed thresholds are used in the separation of the ice classes.

Agreed. We have looked at the sensitivity of the scatterometer class extents to a fixed threshold uncertainty of 0.1 dB (associated to calibration accuracy). The results are plotted in Figure 12.

[Figure]

**Figure 12: Time series of monthly wintertime (March) total multiyear sea ice extents (segmented into SY and older MY classes) within the Arctic Basin from the scatterometer (black) and the NSIDC sea ice age (red) records. Error bars are representative of the scatterometer class extent errors associated to a fixed backscatter threshold uncertainty of 0.1 dB (i.e. calibration accuracy).**

2. Could the backscatter signature of SY ice just be a mixture of MY and FY-ice at the regional transition between the regions with the two dominant ice types?

Indeed, there is potential ambiguity there, particularly before 2007, which is problematic. It is the fact that SY ice appears as a separate mode in the backscatter histograms after 2007 (and not just as a transition between the FY and old MY clusters), along with the fact that it shows a spatial distribution similar to that provided by the NSIDC SY ice class, that encourages us to propose it as a separate entity.

We mention this uncertainty in Section 3.2: "From the analysis of joint backscatter distributions, we know that the scatterometer SY class is bound to contain varying amounts of FY-SY-MY mixtures, and probably some deformed FY too, thus an inherent ambiguity remains regarding the dominance of pure SY ice versus mixed FY-MY combinations in a cell labeled SY, particularly before 2007."

And address it again further below in Section 3.2: "The differentiation of SY and lower concentration of MY using a single frequency remains an open question though. By construction, the scatterometer SY class will accommodate various fractions of deformed FY and FY-SY-MY mixtures in it, which we suggest may be differentiated from the homogenous SY ice signature by recourse to dual Ku-band and C-band observations."

3. The authors' statement that: '...we find relative good agreement between the scatterometer SY ice class and the 2.0 m isoline from the ice thickness record, suggesting the utilization of the backscatter record as a reliable proxy for the estimation of thick sea ice thickness in the Arctic...' is not really supported by the analysis provided here. To demonstrate that the backscatter record is a proxy of thickness would entail more work.

For example, as an assessment, the authors could examine the mean ice thickness of the three ice types over the CryoSat-2 period and then examine the variability within the different categories. In any case, if the authors were to clarify what they meant by 'proxy' it would be more satisfactory.

Certainly. At this point, we cannot demonstrate but only suggest that the backscatter record should be tried as a proxy for the estimation of thick ice thickness (same as Tschudi et al., 2016, http://www.mdpi.com/2072-4292/8/6/457, using sea ice age). Further work along the lines suggested is currently under progress, but is considered out of the scope of this manuscript.

I think these issues should be addressed.

**Reviewer 2**

General comments:

The manuscript "A scatterometer record of sea ice extents and backscatter: 1992–2016" by Maria Belmonte Rivas et al.(tc-2018-68) introduces a data record of sea ice extent and backscatter merging different space-borne scatterometers (ERS, QuikSCAT, and ASCAT). In addition to presenting the methodology and resulting data record, the authors use the new almost 25 years long data to describe recent changes of Arctic sea ice observed after the September 2007 minimum. The contribution is generally of fair quality, and the text is understandable. The paper could be improved by giving some more details on the methodology, and discuss uncertainty of the retrievals.

A major concern are the shortcuts taken when comparing the capabilities of scatterometers (this study) versus (passive microwave) radiometers (e.g. NSIDC and OSISAF SIC data records), especially when it comes to summer melt. The claims of the authors on that specific topic are too often not supported by facts. I encourage the authors to revise the way their findings are presented (not the methodology they apply to compute the data records) and to consider the comments below as an incentive to enhance the quality of their contribution. They should strive at being more balanced, and avoid shortcomings in the presentation of their results.

As a general comment: all the plots (especially the maps) are too small.

Thanks for the detailed revision of the manuscript and the many helpful suggestions, which we have strived to accommodate. We are confident in the superior sensitivity of the scatterometer sea ice extents to melting sea ice conditions, though the reviewer is correct in pointing out that melt ponding is neither the only nor the main reason behind SIE differences. We have certainly not investigated the relationship in detail, so the manuscript is arranged to clarify this point. Please see our replies below.

Specific comments:

1) Abstract: "providing a means to correct for summer melt ponding errors". This is an overstatement. Unless you include in your study an investigation of the melt-pond season (May-June-July), for example using independent melt-pond fraction information (as done in Kern et al. 2017), the statement is not supported. You should be cautious with any statements about the alleged superiority of scatterometers to measure sea ice under melt-ponding conditions : the radar backscatter signal will be very much influenced by the surface water, and by the melting snow. In these conditions, it is hard to believe that scatterometers will have the accuracy to partition the 0%-30% reduced ice concentration (as measured by passive microwave) into some percents of pure "open water" contribution and some percents of pure "melt water" contribution. Such a partition however is what it would take to "provide a means to correct for summer melt ponding errors".

Agreed. The sentence "providing a means to correct for summer melt ponding errors" has been removed. While we provide enough evidence (via validation against MODIS and SAR plates, to be found in the references) to support the superiority of scatterometers to measure sea ice under melting ice conditions, we do not provide any direct supporting evidence (yet?) of its relation to melt-ponding. A more extended commentary is provided as a reply to comment (22).

2) page 1 line 29: "instances of missing thin ice" . . . from where is it missing? from the passive microwave records?

→ "but the precedent scatterometer records also feature instances of missing thin ice during the growth season"

3) page 2 line 5: when comparing sea ice estimates to operational ice charts, especially during summer, one should always question the accuracy of the charts.

Here we are only referring to results obtained by other groups. The validation of satellite sea ice extents admits a very limited set of choices, and operational ice charts are at this point one of the strongest. Direct validation against unambiguous SAR and MODIS scenes is probably better, though more tedious and limited in space and time. We understand the reviewer's concern very well, so we make an effort to accommodate his/her comment:

→ "The validation of the summer sea ice extents from blended records against operational sea ice charts, whose accuracy during summer may also be arguable, shows negative biases by up to 30% (Aaboe et al., 2016), indicating that the distinct sea ice detection skills of scatterometer data may be lost in the blend."

4) page 2 line 14 : "0.1 dB via buoy collocation" (do you mean "via collocation of wind retrievals at ocean buoy locations"?)

Agreed. The paragraph has been modified as suggested.

5) page 2 line 16 and 17: "which are known to cause discontinuities...". please consider the following wording: "which -if not done properly- can affect long-term trends in sea ice concentration". It is fortunate that the intercalibration of brightness temperatures is generally well taken care of by expert teams prior to computing data records of sea ice concentration. Titchner and Rayner (2014) discuss the stitching of SIC from different sources (passive microwave, navigational ice charts, etc..) and not the impact of non-optimal inter-calibration of passive microwave sensors can have on SIC trends.

Agreed. The paragraph has been modified as suggested.

6) page 2 offers a good occasion to re-state that passive microwave records are primarily aiming at mapping sea ice concentration, and sea ice extent is not a goal as such. Sea ice extent is only a downstream indicator.

This comment is accommodated:

→ "Note though, that the primary aim of passive microwave records is the mapping of sea ice concentration, sea ice extent being only a downstream indicator."

7) page 2 line 21: "It is known that...". Then please rather add a good citation.

Added a new citation:

Ulaby, F.T., Moore, R.K., and Fung, A.K.: Microwave Remote Sensing: Active and Passive, Volume III: From Theory To Applications, Artech House Publishers, London, UK, 1981.

8) page 3 line 4: when referring to the ice age data from NSIDC, a more recent citation is 10.1109/JSTARS.2010.2048305.

We added a new citation:

Tschudi, M.A., Fowler, C, Maslanik, J.A., Stroeve, J. 2010. Tracking the movement and changing surface characteristics of Arctic sea ice. *IEEE J. Selected Topics in Earth Obs. And Rem. Sens.*, 10.1109/JSTARS.2010.2048305.

9) page 3, line 20: which Metop platform(s) are used for ASCAT? please specify in the text.

→ "the Advanced Scatterometer (ASCAT) on Metop-A"

10) page 3 line 25: It would be useful to mention here the zenith angles of the different instrument (for example include here the first sentence of section 2.4).

Done – the first sentence of section 2.4 has been moved here.

11) page 3 line 25 : is it worth briefly mentioning NSCAT and why it was not selected in your data record?

NSCAT is a shorter-lived mission. While interesting in that it overlaps the ERS record, featuring collocated Ku and C-band data, its inclusion is envisioned for future extensions of the scatterometer record, along with the newer scatterometer data from ASCAT-B, ASCAT-C, OSCAT, Scatsat, HY, etc…

12) page 5, line 5: "stable wintertime backscatter levels" I understand this as your ice GMF is static, and not dynamically updated (e.g. with months). If I misunderstood, please consider making the sentence more explicit.

That is correct, the ice GMF is static, not dynamically updated.

13) page 6, line 13: please use one sentence to explain the difference in degrees of freedom.

Inserted a new sentence:

→ "The number of degrees of freedom is given by the difference between the size of the measurement space (N, or the number of looks provided by the instrument) and the size of the subspace occupied by backscatter points of a given class, allowing for a two-dimensional ocean wind GMF (wind cone) and a one-dimensional sea ice GMF (sea ice line)."

14) page 6, eq (7) : this is presented as the formula to compute daily estimates of the probabilities, thus supposedly gathering several passess of a given instrument. Yet, equation 7 does not show any indices. Could you make the indices appear in eq 7? Are the individual probabilities multiplied or averaged together?

The details are described in the references (Belmonte Rivas and Stoffelen, 2011; Belmonte Rivas et al., 2012; Otosaka et al., 2017). For completeness, we have inserted the sentence:

→ The *a priori* probabilities are updated at every pass using the previous pass posteriors as $p_0(ice) = p(ice|\sigma) = 1 - p_0(ocean)$, and relaxed towards uncertainty once a day.

15) page 6: it would be very interesting for the users to extend section 2.3 with some material to describe what would be your approach to *providing quantitative uncertainties*. A key discussion point might be how to treat that the PDFs of individual observations might be correlated with each others. To the least, underlined{uncertainties in the retrievals and how to quantify them} should be discussed at the end of the manuscript.

Interesting question. The Bayesian probability already quantifies an uncertainty, whether a pixel belongs to the category of ocean winds, defined by the ocean GMF and its variance, or to the category of sea ice, defined by the sea ice GMF and its variance. In our opinion, quantifying the errors in the Bayesian inference is best left to validation against independent data (passive microwaves, MODIS and SAR plates, sea ice charts).

16) page 6: two informations are missing from section 2: 1) how do you define your SIE (from daily maps of sea ice probability... what is the threshold on probability?) and 2) describe in what sense "these algorithms have been tuned to match the passive microwave sea ice extents during the fall and winter months" (ref your page 2, line 11).

Thanks. We have inserted a new paragraph at the end of Section 2.3:

→ "Our Bayesian approach affords two parameters for tuning: one is the tolerance factor $C_{mix}$ introduced in the sea ice model variance in Eq. (4), and the other is the probability threshold applied on the posterior in Eq. (7). With Quikscat, the tuning parameters have been adjusted empirically to match the passive microwave extents during the fall and winter months, and validated against an extensive series of SAR and MODIS plates during the spring and

summer months (as described in Belmonte Rivas and Stoffelen, 2011), resulting in $C_{mix}$ =3 and a 55% probability threshold to posterior sea ice probabilities. The Bayesian parameters for the ASCAT and ERS configurations have been adjusted similarly, and forced to remain consistent to the Quikscat extents across the mission overlaps periods, resulting in a seasonally varying $C_{mix}$ for ASCAT with a 55% probability threshold, and a seasonally varying $C_{mix}$ with a seasonally varying probability threshold ranging from 40% to 50% for ERS."

17) page 8 lines 9, 14, 15, 18: the use of "biases" is problematic as it implies you consider one source (the scatterometer?) is correct and the other (the passive microwave) is at an offset. Replace with "differences" at all 4 occasions.

All right. The paragraph has been modified as suggested.

18) page 8, lines 7-8 "the agreement... is of comparable high quality during the freezing season"... is it surprising? "these algorithms have been tuned to match the passive microwave sea ice extents during the fall and winter months" (ref your page 2, line 11).

A bit surprising, the tuning of the Bayesian algorithms (only using two parameters, $C_{mix}$ and the probability threshold) seems to work very well.

19) Figure 7: are the colorbars for SIC correct? They would indicate that dark blue is for everything below 20%? It would be interesting to show contour lines of the sea ice extent from NSIDC-0051, OSI-430, and SCATT (e.g. on top of the NIC chart), as this is impossible to observe from the colored maps of SICs.

The colorbars for SIC are correct. The lower limits are 15% SIC for NSIDC-0051 and OSISAF-450 (top panels), and 55% probability for ASCAT (lower left panel). A comment regarding the lower limits has been inserted in the caption of Figure 7. The contour lines of the sea ice extent from NSIDC-0051, OSI-430 and SCAT have been overplotted on the NIC chart.

20) Concerning Figure 7: being an operational product meant for tactical navigation in the ice, the NIC ice charts are not a reference for accurate SIC monitoring. It uses active microwave data (SAR) that might suffer from exactly the same noise sources than the scatterometers, namely that scattered, faceted pieces of ice will seem brighter (higher backscatter) than if the corresponding area of ocean was covered by contiguous ice. Always question the accuracy of NIC charts (or other navigational ice charts), especially during summer.

Thank you for your comment. We are well aware of the ambiguities that make SAR image interpretation difficult. We are not so much concerned about the SIC accuracy of the NIC charts (SIC categories are very coarsely distributed in the NIC charts anyway), as with the fact that they also seem to capture more summer ice than passive microwaves (OSISAF or NSIDC) in a consistent manner.

21) Page 10: "surface wetness and melt ponding...during spring and summer [Comiso,...]" 1) consider also citing Kern et al. (2017) https://doi.org/10.5194/tc-

10-2217-2016 as a more recent and quantitative assessment of the impact of melt-ponds  on the passive microwave retrievals of SIC.

Agreed, added said citation.

22) Concerning Figure 6: it shows a mismatch between PMR and SCATT SIE for both hemispheres and the whole of spring+summer seasons, all the way until the SIE minimum. The sentences following Figure 6 (and Figure 7) point at "surface wetness and melt ponding" as a reason for the mismatch. This is not very intuitive, since:

1) the PMR SIE is defined as the area where SIC is larger than 15%. Although the impact of melt-ponding can be up to 20%-30% (Kern et al. 2017) at the maximum of the season, most of the melt-pond covered area will still show SIC>15%, thus only marginally influencing the PMR SIE.

   It is not unreasonable to think that melt ponding also affects the marginal sea ice.

2) Regardless of item 1), melt-ponding (on sea ice) is happening mostly in (late) May, June, and July, (early August) before they drain through the ice. There are no melt-ponds on top of sea ice in mid September (Figure 7). In addition, melt-ponds on top of sea ice are mostly (if not only) observed in the Arctic. Melt-ponds can thus not explain the SH mismatch you document.

   [Rosel, Kaleschke and Birnbaum, TC, doi:10.5194/tc-6-431-2012] also show remarkable melt pond fractions in the Arctic in early September (see their Fig. 4). Certainly, we cannot say to what extent the SCAT to PMR SIE differences can be exclusively attributed to melt-ponding, as we provide no direct evidence to support a physical relationship - which nevertheless seems plausible.

However:

1) it is noted that the SCATT SIE has a finer spatial resolution than the SSM/I and SSMIS PMR SIE (this is obvious on Figure 7) due to the size of the FoVs of SSMIS at 19 and 37 GHz. Spatial resolution has been documented to have a definite influence on the SIE metric (Notz, 2014, https://www.the-cryosphere.net/8/229/2014/). Can spatial resolution have an influence on the SCATT SIE?

   Generally, a higher grid resolution gives a lower sea-ice extent (Notz, 2014, https://www.the-cryosphere.net/8/229/2014/). So this effect is not likely to have much influence on the summer differences observed (with higher resolution scatterometers observing a larger sea ice extent than passive microwaves).

2) In the marginal ice zone, geometric scattering effects by disjoint, faceted ice floes will induce higher backscatter, and thus artificially enhanced

radar "brightness" than if the same quantity of sea ice was present in a planar, contiguous way. The PMR signal is insensitive to these geometric effects. Could this geometric effect explain the difference in SIE you observe on Figure 6?

In our experience, most of the spring and summer errors along the ice margin (both in the NH and SH) involve darker surfaces such as shown in Figure 9B in (Belmonte Rivas and Stoffelen, 2011), which is suggestive of water saturated ice. For a more extended collection of evidence illustrating the nature of the SCAT to PMR summer differences using SAR and MODIS imagery, the reader is referred to the OSISAF visiting scientist report:

https://cdn.knmi.nl/system/data_center_publications/files/000/068/084/original/sea_ice_osi_saf_final_report.pdf?1495621021

3) Finally, if indeed your sea ice GMF is static and tuned for winter conditions (see page 5, line 5), then you should discuss if the spring+summer sea ice GMF is similar (than the winter one) so that to ensure that seasonality of the sea ice GMF does not artificially contribute to seasonality in SCATT SIE.

Back in the initial development stages, we did look at the seasonality in the distribution of sea ice backscatter measurements. The mode of the sea ice GMF remains essentially the same across the seasons, but in the spring and summer months, a large cloud of points is drawn towards the ocean GMF to form an extended tail of mixed sea ice and open ocean conditions. We did take care of this tail by introducing the tolerance $C_{mix}$ parameter in Eq (4).

It is the (different degree of) inclusion of these mixed sea ice and open ocean conditions that is mainly responsible for the SCAT to PMR differences in the spring and summer months. Upon validation against SAR and MODIS plates (see the OSI-SAF report mentioned above), we learned that these mixed conditions include a variety of scenes, including low concentration ice (decaying sea ice floes), ice bands, brash (water saturated) ice, and mixtures thereof.

All in all, and as noted in the introduction to this review: the readability and impact of this paper would be greatly improved if 1) you described the observed differences between PMR and SCATT SIE without too hastily attributing them to deficiencies of the PMR SIEs (specifically melt-ponding), 2) you investigated (or pointed to earlier investigations) how other factors might explain (better?) the differences.

We provide ample evidence (in the references) to support the fact that observed differences between PMR and SCAT SIE indicate that SCAT is more sensitive to mixed sea ice and open ocean conditions (particularly prevalent in the spring and summer months) than PMR. The reviewer is correct in pointing out that melt

ponding is neither the only nor the main reason behind SIE differences. We have certainly not investigated the relationship in detail, so the manuscript is arranged to clarify this point.

The paragraph (page 10, line 1) has been modified:

→"Surface wetness and melt ponding are thought to be responsible for large errors in passive microwave sea ice concentrations during spring and summer (Comiso and Kwok, 1996) (Kern et al, 2016), and these errors affect the ocean heat contents and associated surface fluxes when assimilated into ocean and atmosphere reanalyses (Hirahara et al., 2016). In this context, the scatterometer record nicely complements the passive microwave products in monitoring  the expanse and evolution of the _lower concentration and water-saturated_ (rotten) late spring and summer sea ice classes. _It is the different degree of inclusion of these mixed sea ice and open ocean conditions that is mainly responsible for the sea ice extent differences observed between scatterometers and passive microwaves in the spring and summer months. The reader is referred to (OSISAF Visiting Scientist Report) for a more extended collection of collocated SAR and MODIS plates illustrating the nature of the differences observed in Figs. 5-6, which include a variety of scenes with decaying sea ice floes, ice bands, water saturated (brash) ice, and mixtures thereof._"

OSISAF Visiting Scientist Report:
https://cdn.knmi.nl/system/data_center_publications/files/000/068/084/original/sea_ice_osi_saf_final_report.pdf?1495621021

In the abstract (page1, line 12):

→ "… but shows higher sensitivity to lower concentration and melting sea ice during the spring and summer months."

In the conclusions (page 18, line 21)

→"In this context, the scatterometer sea ice extents and probabilities nicely complement the passive microwave products in providing a basis to monitor the occurrence of sea ice concentration errors due to  surface wetness, and to delineate the expanse and evolution of the rotten late spring and summer ice classes."

23) Figure 11: a legend/colorbar is missing. In addition the maps are too small (this applies to all figures in the draft manuscript). Figure 11 shows large discrepancies between the left and middle panels outside the central arctic ocean (e.g. second year ice in Bering Sea in 2008 for ASCAT but not QSCAT). Please either zoom your maps to the Arctic Basin (this is the area you discuss anyway, or comment the large discrepancies (how they can be mitigated).

Figure 11 has been enlarged, and a colorbar has been added.

For clarity, we introduced the following sentence:

→ "Outside of the red contour that delineates the Arctic basin mask in the left and middle panels of Figure 11, we cannot register older ice reliably because of the strong backscatter from deformed MIZ ice."

24) Page 15, line 18: as noted later in the text, the OSI-SAF product OSI-403 does not hold a MY ice concentration, but a MY/FY classification.

That is correct. The sentence is modified:

→ "The monthly averaged MY ice concentration for the month of March 2016 is calculated directly over the daily MY ice concentrations from the SICCI and Bremen products. Note that the OSISAF-403 is not a sea ice concentration product but a FY/MY classification. In this case, a daily MY concentration is defined (100% for the MY class, 50% for the ambiguous class, and 0% for the FY or OW classes) and a monthly average MY concentration is calculated as above. The monthly averaged sea ice age is calculated over the weekly NSIDC grids (using weeks 9 to 12) and over the daily SICCI grids, defining the SY ice class for a monthly averaged sea ice age between 1.5 and 2.5 years."

25) Figure 13: Several remarks:

1) the panels are too small.

   Now enlarged.

2) what date is this from (inside March 2016)? Or is this a monthly average for all panels? If a monthly average you should probably describe how the average is performed, given the variety of input variables (MYI conc, MYI classification, max SIA, mean SIA,...). It would make more sense to have a specific date.

   It is a monthly average, as stated in the caption. See our reply to comment (24) above.

3) consider adding coast lines.

   A coastal mask is already applied.

4) legend : the OSI-403 is not a MYI concentration product but a sea ice classification.

   See our reply to comment (24) above.

5) please add a "first year ice" color (or a sea ice edge line) on panels b) to f).

Thanks for the suggestion. This figure discusses the representation multiyear ice. Adding a sea ice edge for FY would not help this purpose.

6) please check your plot c) and f) (from Korosov et al. 2017 data). If this is plotted from variable "sia" (the mean sea ice age), one has to look at 1<sia<2 for the second year ice (as soon as sia is > 1, then it has 2nd year ice contribution). The result will look much more similar to NSIDC SIA and the other maps, especially in the Beaufort Sea.

We checked the plots after redefining the SY class using a monthly averaged SIA between 1 and 2 years, as suggested. The problem with the apparently missing SY ice over the Beaufort Sea in the SICCI product is not related to the SIA, but related to the MYI concentration, which is lower than 30% - thus masked in the figure.

7) the description of the location of the features is difficult to follow in the text, would it be an idea to write letters on the map (A, B, C, ...) and refer to them?

Agreed and done.

8) all estimates (but yours in a) show a thin tongue of older ice extending across the Arctic Ocean towards the New Siberian Islands. It can even be noticed somewhat in the Cryosat-2 thickness. It is visible on the shades of "normalized backscatter" in your panel a). Given the variety of methods used for all other panels, are you confident that your new estimate in a) is correct and that there is no such tongue of older ice? Discuss.

Good point. We pondered over this feature at length, and it does justice to mention it.

→ "Another interesting feature refers to the thin tongue of older ice extending across the Arctic Basin towards the New Siberian Islands (see label D in Fig. 13g), which is seen by all products, even faintly in the AWI sea ice thickness, but falls below the SY threshold in the scatterometer-based MY ice classification. We cannot offer an explanation for this feature at the moment, other than acknowledging that efforts towards ensuring the consistency among MY ice products in the Arctic should warrant further research."

26) Page 17, line 15 : "as verified by comparison to NIC sea ice charts". NIC sea ice charts are not the truth. Please use a different wording than "verify"..

→ "… but show enhanced sensitivity to lower concentration and water-saturated sea ice conditions during the spring and summer months, as verified by numerous comparisons to MODIS and SAR imagery."

27) Page 17, line 18 : "typically underestimating the summer sea ice concentration and summer sea ice extent by up to 30%". Which one? SIC or SIE?

As discussed above an underestimation of high SICs, even by 30% will have limited impact on the SIE (defined as SIC>15%). Rewrite

We have cited a number of references reporting estimates of PMR SIC and SIE errors up to 30% in the summer. Another comparison (OSISAF 403 against NIC charts in the SH) here:

http://osisaf.met.no/validation/img/sh_edge_bar_plot_full.png

28) page 17, line 21: "providing a solid basis to monitor the occurrence of sea ice concentration errors due to melt ponding". NO. See notes from the introduction. Rewrite.

Agreed. This sentence has been removed; please see our reply to comment (22) above.

29) page 17, line 29, 30: this sentence is an exact copy-paste from previous section. Avoid word-by-word repeats, please.

Thanks for the suggestion. This sentence has been reworded into:

→ "Monitoring the evolution of the complementary old MY and SY classes shows that the decline in the total MYI ice extent observed in the Arctic has been driven by loss of old MY ice, while the more steady production of SY ice has been acting to stabilize those losses, contributing to later recovery events such as observed in 2014."

30) page 18, line 15 : to use Sea Ice Type (or Age) as a proxy for Sea Ice Thickness is not a new idea. It would be good to cite earlier attempts e.g. Tschudi et al. 2016, http://www.mdpi.com/2072-4292/8/6/457.

Thanks for the suggestion. A new citation has been inserted (page 16, line 10):

Tschudi, M.A., Stroeve, J. and Stewart, J.S.: Relating the age of Arctic sea ice to its thickness, as measured during NASA's ICEsat and IceBridge, *Remote Sens.* **2016**, *8*(6), 457; https://doi.org/10.3390/rs8060457 , 2016.

Technical corrections:

31) page 1 line 14: "record sea ice loss" : consider another term than "record" as it is used elsewhere in the paper (including the abstract) as "data record".

→ "historical sea ice loss"

32) page 1 line 19: "1978" (with the start of SMMR in Oct 1978). There are passive microwave instruments before (e.g. ESMR), but the routine monitoring indeed starts in 1978 with SMMR.

→ "1978"

33) page 3 line 22: "mission transition periods".. consider "mission overlap periods"

→ "mission overlap periods"

34) page 3, line 13; "measurements about extended"...do you mean "around"?

→ "measurements around extended"

35) Figure 4: please add a second legend with solid line for NH and dashed line for SH.

Done.

36) Figure 5: please add text in the plot area for NH (top) and SH (bottom).

Done.

37) page 7, line 15 : you are probably referring to OSI-430, that extends OSI-409a (http://osisaf.met.no/p/ice/index.html#conc-reproc). Same in caption to Figure 7.

That is correct, OSI-430. We have amended the typo, both in text and caption.

38) page 8, line 9 : "Quikscat-to-ERS"... you probably mean "ERS-Quikscat"?

That is correct → "ERS-to-Quikscat"

39) Figure 6: you seem to have a dip in QuikSCAT SIE in antarctic curve for 2000 (around day 260).

It is a dip in the Antarctic OSISAF0-409a SIE curve. We checked the daily OSISAF 409a maps around the suggested date, but did not find anything strange (i.e. it does not seem to be an instrumental artifact…)

40) page 10, line 24: "arrival of summer signatures" consider "appearance of summer signatures"?

→ "appearance of summer signatures"

---

## Author Response (AR2)

**Response to reviews:**

Our thanks go to the handling editor and the two anonymous reviewers for the time they have dedicated to reading and helping us improve the manuscript. A short reply to the latest correction suggested by Reviewer #2 is appended below.

Reviewer #1

[Accept as is - No comments]

Reviewer #2

[The revised version of your manuscript is greatly improved, and I fully support publication in EGU TC. The text is now more open about the challenges the scatterometer record has (e.g. the discussion about label D on Figure 13). The methodology is better documented, with due reference to earlier work. The text is also more balanced wrt to the passive microwave sea ice concentration/extent records, and what can be expected from navigational ice charts.

I just have a final technical suggestion to improve the readability of your maps, for example Figure 7, bottom right panel, the OSISAF 15% contour is hard to see.]

We have modified the color table in the bottom right panel of Fig. 7, in order to improve the readability of the OSISAF 15% sea ice contour:

[Figure]

Former           →           Final Version

It is difficult to plot so much data on a single panel, but we think that the final version is clearer. Thanks again and good luck.